# Encapsulation of *Cymbopogon khasiana* × *Cymbopogon pendulus* Essential Oil (CKP-25) in Chitosan Nanoemulsion as a Green and Novel Strategy for Mitigation of Fungal Association and Aflatoxin B_1_ Contamination in Food System

**DOI:** 10.3390/foods12040722

**Published:** 2023-02-07

**Authors:** Jitendra Prasad, Somenath Das, Akash Maurya, Monisha Soni, Arati Yadav, Bikarma Singh, Abhishek Kumar Dwivedy

**Affiliations:** 1Laboratory of Herbal Pesticides, Centre of Advanced Study (CAS) in Botany, Banaras Hindu University, Varanasi 221005, Uttar Pradesh, India; 2Department of Botany, Burdwan Raj College, Purba Bardhaman 713104, West Bengal, India; 3Botanic Garden Division, CSIR-National Botanical Research Institute, Rana Pratap Marg, Lucknow 226001, Uttar Pradesh, India

**Keywords:** CKP-25-essential oil, aflatoxin B_1_, mitigation, preservative, methylglyoxal, molecular docking

## Abstract

The present study deals with the encapsulation of *Cymbopogon khasiana* × *Cymbopogon pendulus* essential oil (CKP-25-EO) into a chitosan nanoemulsion and efficacy assessment for inhibition of fungal inhabitation and aflatoxin B_1_ (AFB_1_) contamination in *Syzygium cumini* seeds with emphasis on cellular and molecular mechanism of action. DLS, AFM, SEM, FTIR, and XRD analyses revealed the encapsulation of CKP-25-EO in chitosan with controlled delivery. The CKP-25-Ne displayed enhanced antifungal (0.08 µL/mL), antiaflatoxigenic (0.07 µL/mL), and antioxidant activities (IC_50 DPPH_ = 6.94 µL/mL, IC_50 ABTS_ = 5.40 µL/mL) in comparison to the free EO. Impediment in cellular ergosterol, methylglyoxal biosynthesis, and in silico molecular modeling of CKP-25-Ne validated the cellular and molecular mechanism of antifungal and antiaflatoxigenic activity. The CKP-25-Ne showed in situ efficacy for inhibition of lipid peroxidation and AFB_1_ secretion in stored *S. cumini* seeds without altering the sensory profile. Moreover, the higher mammalian safety profile strengthens the application of CKP-25-Ne as a safe green nano-preservative against fungal association, and hazardous AFB_1_ contamination in food, agriculture, and pharmaceutical industries.

## 1. Introduction

*Syzygium cumini* L. (also known as Jamun), belonging to the family Myrtaceae, is a commonly cultivated medicinal plant in tropical and subtropical regions throughout the world [1]. The seeds are beneficial for the treatment of diabetes and inflammatory disorders due to presence of phenolic and bioactive components like carotenoids, ellagic acid, tannic acid, caffeic acid, quercetin, and polyphenols [1,2]. During the storage conditions, *S. cumini* seeds are infested by a variety of fungi such as *Aspergillus*, *Penicillium*, and *Fusarium*, resulting in excessive loss of medicinal properties and nutrient components due to the production of hazardous mycotoxins, and thus significantly reducing their market value [3]. Among different reported mycotoxins, aflatoxin B_1_ (AFB_1_) produced by *Aspergillus flavus* has been paid serious attention due to their hazardous impact on human health and animals [4]. Most importantly, AFB_1_ is considered as a carcinogenic, teratogenic, mutagenic, and immunosuppressive agent, hence, classified as Class I human carcinogen by International Agency for Research in Cancer [5]. In addition, oxidative stress-mediated lipid peroxidation has a negative impact on stored food commodities after AFB_1_ contamination, causing off-flavor, off-odor, and deterioration of nutrient components [6]. Moreover, free radicals stimulate the biosynthesis of cellular methylglyoxal and have been reported as an aflatoxin influencing molecule in *A. flavus* [7]. Varieties of synthetic preservatives are being used to combat fungi and aflatoxin contamination in food commodities. However, regular use of these chemical preservatives may develop resistance in target organisms and pose harmful effects to human health and the environment [8]. In this context, essential oils and their bioactive components have been used for inhibition of fungi and AFB_1_ contamination due to their friendly nature to ecosystem, and inclusion under the Generally Recognized as Safe (GRAS) category [9]. However, practical applications of essential oils as food preservatives have some limitations based on their highly volatile nature, low water solubility, sensitivity to light, oxygen, and chances for alteration in organoleptic properties of food commodities [10]. These challenges of essential oils are ameliorated by innovative nanoencapsulation technologies where the stability of essential oils is retained into any compatible polymer matrix [11]. By using this method, essential oils have been encapsulated as a core material inside a polymeric wall matrix with long-term preservation of chemical properties [12]. A number of natural and synthetic polymers like chitosan, starch, alginate, polyamines, polylactate, and polyglycolate have been used as a wall material during nanoencapsulation; among them, chitosan (cationic polysaccharide and derived from deacetylation of chitin) are an ideal agent for encapsulation of essential oils and their bioactive components due to their biodegradability, biocompatibility, cost-effectivity, antimicrobial nature, and non-toxic properties [13]. Chitosan-based encapsulation of essential oil is an effective approach to facilitate agricultural application with significant entrapment of bioactive constituents [14]. Most importantly, the chitosan-based nanocarriers present the promising opportunity for controlled delivery of essential oil with outstanding performance in inhibition of fungal and microbial deterioration of foods [15]. The encapsulation of essential oil into chitosan is based on the aptitude of chitosan for aggregation in contact with water while the hydrophobic essential oil is sequestrated at the centre by some carboxyl, aldehyde, and amine functional groups which help in sustained delivery for longer time duration—a primordial importance as nano-green preservative in agricultural sectors [16]. Varieties of nanoencapsulation technology have been employed for entrapment of essential oils, however, ionic gelation-based homogenization and the sonication method with resultant formation of nanoemulsion are of great significance for the antifungal and mycotoxin mitigation process due to higher stability, control delivery, aqueous phase suitability, and better dispersibility [17].

Essential oils of Lemongrass (*Cymbopogon pendulus* variety CKP-25) are commonly extracted from their leaves and widely used in food, perfumery, cosmetics, and pharmaceutical industries. Lemongrass variety CKP-25 is cultivated in the Indian subcontinent based on high essential oil content with prominent antimicrobial, anti-inflammatory, and anti-diabetic activities [18]. However, there are no reports on antifungal and AFB_1_ inhibitory abilities of CKP-25-EO in stored food commodities with an emphasis on cellular and molecular mechanisms of action and improvement in bioefficacy through novel nanoencapsulation strategy.

Hence, the current investigation aimed to encompass the CKP-25-EO within chitosan nanobiopolymer (CKP-25-Ne) to boost the fungal and AFB_1_ suppressing efficacy in stored *S. cumini* seeds (model food system). The characterization of CKP-25-Ne was conducted through DLS, AFM, SEM, XRD, and FTIR techniques. The effect of CKP-25-Ne on fungal ergosterol, leakage of cellular cations, and methylglyoxal biosynthesis (AFB_1_ inducer) has been associated with antifungal and antiaflatoxigenic mechanisms of action. In silico modeling suggested the molecular mechanism to combat the AFB_1_ synthesis. Moreover, in situ antifungal, AFB_1_ mitigation potentiality, and the lipid peroxidation regulating ability in *S. cumini* seeds—along with favorable safety paradigms—extend the practical utilization of CKP-25-Ne in food, agriculture, and pharmaceutical industries as a smart plant-based preservative.

## 2. Materials and Methods

### 2.1. Reagents and Chemicals

Reagents and chemicals like SMKY (sucrose, potassium sulphate, magnesium sulphate), yeast extract, PDA (potato dextrose agar), toluene, isoamyl alcohol, chitosan (low molecular weight), sodium-tripolyphosphate (S-TPP), Na_2_SO_4_, dichloromethane (DCM), potassium carbonate, diamino benzene, HClO_4_, glacial acetic acid, dimethyl sulfoxide (DMSO), potassium bromide, chloroform, ethyl acetate (EA) (used to dissolve essential oil), methanol (used as solvent), Tween-20, Tween-80 (used as surfactant), thiobarbeturic acid (TBA), trichloroacetic acid (TCA), hydrochloric acid (HCl), 2,2′-azino-bis(3-ethylbenzothiazoline-6-sulfonic acid) (ABTS), methanol, and 2,2-diphenyl-1-picrylhydrazyl (DPPH) were purchased from Hi-Media Laboratory, Mumbai, India.

### 2.2. Test Fungal Strain

Fungal strains of *Aspergillus flavus* (AFLHPSc-1), *Aspergillus niger*, *A. luchuensis*, *A. humicola*, *A. spinulosum*, *A*. *glaucus*, *Penicillium fellutemum*, and *P. albicans,* isolated and identified in our previous investigation, were used as test strain for present study [3].

### 2.3. Extraction of Lemongrass (CKP-25) Essential Oil

The whole plant material of lemongrass (CKP-25) was used for oil extraction using the method of Sawadogo et al. [19] with slight modification. At lab scale, 500 g of sample was used for hydro-distillation using Clevenger’s apparatus at 100 °C and run for 4 h. Purification of essential oil was conducted by removing the insoluble particles and dried overnight by mixing it with anhydrous Na_2_SO_4_.

### 2.4. Chemical Characterization of CKP-25-EO

Chemical characterization of CKP-25-EO was performed through GC-MS (GC-2030, TQ8040NX Shimadzu). SH-Rxi-5 SILMS (0.25 × 30 × 0.25 µm) column was used for the GC-MS investigation; it was used as the carrier gas and was passed through GC with 1 mL/min flow. Initially, the column temperature was set at 70 °C for 10 min, and gradually enhanced to 150° with a 7 °C/min rate. The injector and ion source temperature was 260° and 220°, respectively. Dilution of samples in 10:100 *v*/*v* ethyl acetate and then 2 µL samples was injected through an auto sampler injector. The ionization energy mass range was set up at 70 eV and 40–500 AMU. Then, compounds were identified by using NIST library.

### 2.5. Synthesis of CKP-25-EO Loaded Chitosan Nanoemulsion (CKP-25-Ne)

Encapsulation of CKP-25-EO into the chitosan nanomatrix was performed through the ionic gelation process of Yoksan et al. [20] with minor changes. For this, 1.25 g chitosan was added to 100 mL 1% glacial acetic acid solution and mixed by keeping it at magnetic stirrer overnight. Tween-80 (0.413 mL) was added dropwise in chitosan solution followed by mixing of different volumes of CKP-25-EO (0.06, 0.12, 0.18, 0.24, and 0.30 g) during homogenization at 12,890 rpm for 15 min. After that, 0.38% (*w*/*v*) S-TPP solution was blended into the chitosan solution, and again agitated over the magnetic stirrer for 40 min. After centrifugation, the pellet was washed with double distilled water (DDW) and sonicated for ¼ h (1 min interval) with the resultant formation of nanoemulsion. A similar process was used for preparation of the chitosan nanoemulsion without addition of CKP-25-EO. Finally, the nanoemulsion was dried in lyophilizer to develop into powdered form (nanoparticle). Biological efficacy assessments were performed by using nanoemulsion, whereas, powdered form was used for instrumental characterizations.

### 2.6. Measurement of Efficiency to Load and Encapsulate CKP-25-EO

Briefly, 300 µL of CKP-25-Ne was homogenized properly in 3 mL EA and centrifuged at 8700× *g* rpm for 12 min. The collected supernatant was used for the determination of EE and LC on the basis of standard curve of CKP-25-EO prepared in ethyl acetate with absorption maxima at 291 nm (y = 0.0015x + 0.0231, *R*^2^ = 0.9995). The formula for measurement of EE and LC are presented below.
$$\;\mathrm{Encapsulation}\;\mathrm{efficiency}\;\left(\mathrm{EE}\right)\left(\%\right)=\frac{\mathrm{Total}\;\mathrm{amount}\;\mathrm{of}\;\mathrm{CKP}-25\;\mathrm{into}\;\mathrm{nanoemulsion}}{\mathrm{Initial}\;\mathrm{amount}\;\mathrm{of}\;\mathrm{CKP}-25-\mathrm{EO}}\times 100$$
$$\mathrm{Loading}\;\mathrm{capacity}\;\left(\mathrm{LC}\right)\;\left(\%\right)=\frac{\mathrm{Total}\;\mathrm{amount}\;\mathrm{of}\;\mathrm{CKP}-25-\mathrm{EO}\;\mathrm{loaded}\;\mathrm{in}\;\mathrm{the}\;\mathrm{sample}}{\mathrm{Weight}\;\mathrm{of}\;\mathrm{lyophilized}\;\mathrm{nanoemulsion}}\times 100$$

### 2.7. Characterization of CKP-25-Ne

#### 2.7.1. Dynamic Light Scattering (DLS) Analysis

First, 10 mg lyophilized chitosan nanoemulsion and CKP-25-Ne were diluted in 10 mL distilled water. Subsequently, samples were homogenized for 8 min followed by measuring of average polydispersity index (PDI), zeta potential, and particle size using Zeta sizer (Ultra, Malvern Panalytical, UK).

#### 2.7.2. Morphological Characterization

First, 10 mg powder of chitosan nanoemulsion, and CKP-25-Ne was homogenized in 10 mL of deionized water, sonicated, and one drop of suspension was spread on thin tape succeeded by desiccating in sterile air. Thereafter, gold coated dispersed film was observed in SEM (Evo-18 researcher, Zeiss, India).

#### 2.7.3. Atomic Force Microscopy (AFM) Analysis

In brief, 1 mg dried CKP-25-Ne sample was diluted in 10 mL deionized water and homogenized for ¼ h. Then, 10 microlitres of the diluted suspension was spread on the coverslip to make a thin layer. After that, the sample was dried at room temperature, and observed under AFM (CSM Instrument, Tritec, Pesuex, Switzerland).

#### 2.7.4. Fourier Transform Infrared Spectroscopy (FTIR) Evaluation

The FTIR analysis of CKP-25-EO, chitosan powder, chitosan nanoemulsion, and CKP-25-Ne was conducted by Perkin Elmer Fourier transform infrared spectrometer (USA) in between 500–4500 cm^−1^ with resolution power 4 cm^−1^.

#### 2.7.5. X-ray Diffraction (XRD) Observation

XRD analysis of chitosan powder, chitosan nanoemulsion, and CKP-25-Ne was performed using X-ray diffractometer under 40 mA and 40 kV operating system with Cu-Kα radiation. The sample was scanned at a 2θ value ranging 5–50° with scan speed 5° min^−1^.

### 2.8. Release Profile of CKP-25-Ne

The release characteristic of CKP-25-EO was determined according to the protocol of Hasheminejad et al. [21]. A total of 0.05 mL CKP-25-Ne was homogenized in phosphate buffer saline (PBS) (pH = 7.4) and ethyl alcohol solution (3:2 *v*/*v*) followed by centrifugation at 12,950× *g* rpm for 10 min. At different time intervals, 100 µL of the upper layer was separated in an eppendorf tube, and optical density was noted at 297 nm (standard curve of CKP-25-EO was obtained in PBS + ethanol solution) by UV-Visible spectrophotometer. In all the cases, sink conditions were maintained by fresh PBS. Cumulative release of CKP-25-EO into alcoholic buffer solution was measured by the following formula.
$$\mathrm{Release}\;\mathrm{of}\;\mathrm{CKP}-25-\mathrm{EO}\;\left(\%\right)=\frac{\mathrm{Cumulative}\;\mathrm{release}\;\mathrm{CKP}-25-\mathrm{EO}\;\mathrm{content}\;\mathrm{from}\;\mathrm{nanoemulsion}}{\mathrm{Initial}\;\mathrm{amount}\;\mathrm{of}\;\mathrm{CKP}-25-\mathrm{EO}\;\mathrm{loaded}\;\mathrm{in}\;\mathrm{the}\;\mathrm{sample}}$$

### 2.9. Antifungal and AFB_1_ Inhibitory Efficacy of CKP-25-EO and CKP-25-Ne

Antifungal activity of CKP-25-EO and CKP-25-Ne against AFLHPSc-1 was determined following the method of Das et al. [22]. For determination of antifungal activity, required amounts of CKP-25-EO (0.05–0.4 µL/mL) and CKP-25-Ne (0.01–0.08 µL/mL) were amended into conical flasks containing 25 mL SMKY medium. Two control sets were prepared; 500 µL of 5% Tween-20 was used as control for CKP-25-EO, while chitosan nanoemulsion was used as control for CKP-25-Ne. Thereafter, 25 µL AFLHPSc-1 spore suspension was added in both of the sets and incubated in BOD at 27 ± 2 °C for 10 days. The concentration of CKP-25-EO and CKP-25-Ne at which the growth of AFLHPSc-1 completely disappeared has been regarded as the minimum inhibitory concentration (MIC).

For antiaflatoxigenic assay, the media were transferred in a separating funnel containing 20 mL chloroform. The media were extracted and evaporated at 65 °C for 30 min. Residue was mixed with methanol and was loaded on a thin layer chromatography plate (TLC) and immersed into a solvent solution comprising of Isoamyl alcohol: Toluene: Methanol, 32: 90: 2, *v*/*v*/*v* for 30 min. Light blue spots were examined on UV-transilluminator, separated, and added into 10 mL methanol and centrifuged at 13,000× *g* rpm for 10 min. The upper layer was analyzed at 360 nm by using a UV-Visible spectrophotometer. Finally, the amount of AFB_1_ was measured following the formula of Das et al. [23].

### 2.10. Determination of Fungitoxic Spectrum of CKP-25- EO and CKP-25-Ne

For this, 10 mL PDA medium was introduced in Petri plate and inoculated with 5 mm disc of different fungal species viz. *Aspergillus niger*, *A. luchuensis*, *A. humicola*, *A. spinulosum*, *A*. *glaucus*, *Penicillium fellutemum*, and *P. albicans*, followed by treatment with an MIC dose of CKP-25-EO (0.4 µL/mL) and CKP-25-Ne (0.08 µL/mL). PDA plates with fungal species without any treatment served as control. Thereafter, the control as well as fumigated sets were incubated in BOD at 27 ± 2 °C for 7 days. Mycelial growth inhibition was calculated by subtracting the treatment set fungal diameter from the control set fungal diameter.

### 2.11. Action Mode Related to CKP-25-EO and CKP-25-Ne

#### 2.11.1. Effect of CKP-25-EO and CKP-25-Ne on Ergosterol

Briefly, 25 mL SMKY media was poured in conical flask and 25 µL AFLHPSc-1 spore suspension was added. After that, fumigation of AFLHPSc-1 cells was performed with different concentrations of CKP-25-EO (0.05–0.4 µL/mL) and CKP-25-Ne (0.01–0.08 µL/L). Control sets were prepared without fumigation of CKP-25-EO and CKP-25-Ne. The control and fumigated sets were incubated at 27 ± 2 °C for 7 days. After that, the mycelia were weighed and homogenized in 5 mL alcoholic KOH (25%) solution with heating (65 °C, ½ h). For extraction of ergosterol, 2:5 mL (*v*/*v*) aqueous n-heptane solution was added to the mycelial sample. The upper n-heptane layer was used for analysis of the ergosterol content being between 230–300 nm using UV-Visible spectrophotometer. The ergosterol content was calculated by using the formula given by Tian et al. [8].

#### 2.11.2. Effect of CKP-25-EO and CKP-25-Ne on Cellular Constituents Leakage

For this, five-day-old mycelia of AFLHPSc-1 were harvested, washed, and treated with different concentrations (1/2 MIC, MIC, and 2 MIC) of CKP-25-EO and CKP-25-Ne in 20 mL saline solution (0.85% *w*/*v*) and incubated at ambient temperature overnight. Control sets were devoid of CKP-25-EO and CKP-25-Ne. After the incubation period, filtrates were filtered by filter paper and then subjected for measurement of leakage of ions (Ca^2+^, Mg^2+^, and K^+^) by using an Atomic Absorption Spectrophotometer (AAS) and 260, 280 nm absorbing materials by UV-Visible spectrophotometer [24].

#### 2.11.3. Effect of CKP-25-EO and CKP-25-Ne on Cellular Methylglyoxal (AFB_1_ Inducer)

Seven-day-old AFLHPSc-1 mycelial biomass was harvested, washed, and 0.3 g of biomass was transferred to 10 mL SMKY medium. Thereafter, the mycelial biomass was fumigated with different concentrations (1/2 MIC, MIC, and 2 MIC) of CKP-25-EO and CKP-25-Ne, succeeding by keeping in BOD at 27 ± 2 °C overnight. Control sets were devoid of CKP-25-EO and CKP-25-Ne. After overnight incubation, biomass was homogenized into perchloric acid (0.5 M) and centrifuged at 10,000× *g* rpm for 14 min. The upper layer was neutralized using saturated potassium carbonate and again centrifuged at 13,000× *g* rpm for 10 min at 4 °C. For methylglyoxal estimation, 0.25 mL of 1,2 diaminobenzene (7.2 mM) was added to 0.1 mL perchloric acid (5 M). Finally, the total volume was maintained at 1 mL by adding neutralized supernatant. Subsequently, the absorption of the samples was noted at 336 nm, and methylglyoxal content was obtained with the help of the calibration curve of methylglyoxal [7].

#### 2.11.4. Molecular Docking of Citral (Major Component of CKP-25-EO) with Nor-1 and Pks-A Proteins of AFB_1_ Secretion

The 3D structure of the citral was obtained from the PUBCHEM online database in SDF format. SWISS DOCK, Phyre 2, and Uniprot online servers were utilized for the development of 3D structures of nor-1 and pks-A proteins of *Aspergillus flavus*. Binding of citral with nor-1 and pks-A proteins was performed with the help of CHIMERA 1.13. Best interactions of citral with nor-1 and pks-A proteins were managed by using number of hydrogen bonds and binding energy index [22].

### 2.12. Antioxidant Activity and Total Phenolic Content of CKP-25 and CKP-25-Ne

#### 2.12.1. DPPH^·+^ Assay

For this, 2 mL methanolic DPPH (0.004%) solution was mixed with different concentrations of CKP-25-EO and CKP-25-Ne and incubated in the dark for 30 min at room temperature [25]. Subsequently, absorbance measurement was done at 517 nm and calculation of radical quenching potentiality was performed by the following formula. IC_50_ (50% radical neutralizing capacity) was determined by plotting the graph of % radical neutralization against CKP-25-EO and CKP-25-Ne concentrations.
$$\%\;\mathrm{Inhibition}\;=\frac{\mathrm{Absorbance}\;\mathrm{of}\;\mathrm{blank}-\mathrm{Absorbance}\;\mathrm{of}\;\mathrm{sample}\;}{\mathrm{Absorbance}\;\mathrm{of}\;\mathrm{sample}}\times 100$$

#### 2.12.2. ABTS^·+^ Assay

For ABTS^·+^ analysis, 1 mL ABTS^+^ solution (7 mM) was mixed in 1 mL K_2_S_2_O_8_ (14 mM) solution followed by incubation in the dark overnight [26]. After incubation, different concentrations of CKP-25-EO and CKP-25-Ne were added to the ABTS mixture solution, and absorbance was noted at 734 nm through UV-Visible spectrophotometer. Radical quenching potentiality was determined by using the equation already mentioned in Section 2.12.1.

#### 2.12.3. Total Phenolic Content Estimation

For estimation of phenolic content, 10 µL CKP-25-EO and CKP-25-Ne were dissolved in 100 µL DMSO followed by homogenization in 46 mL distilled water. The sample mixture was added with 1 mL Folin ciocalteu reagent. Thereafter, 3 mL Na_2_CO_3_ (2%) solution was mixed, and absorption measurement was performed at 760 nm using a UV-Visible spectrophotometer [3]. Phenolic content was calculated through the following formula.
Absorbance=0.0012×Gallic acid (µg)+0.024

### 2.13. In Situ Antifungal and Anti-AFB_1_ Efficacy of CKP-25-EO and CKP-25-Ne in Stored S. cumini Seeds (The Model Anti-Diabetic Food System)

In brief, 500 g *S. cumini* seeds were stored in a plastic container for one year at 25–30 °C and relative humidity 30–70% under fumigation with CKP-25-EO and CKP-25-Ne at MIC and 2 MIC doses. Control sets were not treated with CKP-25-EO and CKP-25-Ne. After completion of the storage period, 5 g of grinded seed samples (control and treated) were dissolved in 10 mL distilled water and 1000 µL aliquot of dilution of 1/10,000 series was introduced in 10 mL potato dextrose agar medium. The antifungal activity was measured by the following equation.
$$\%\;\mathrm{Antifungal}\;\mathrm{activity}\;=\frac{\mathrm{Fungal}\;\mathrm{contamination}\;\mathrm{in}\;\mathrm{untreated}\;\mathrm{seed}-\mathrm{Fungal}\;\mathrm{contamination}\;\mathrm{in}\;\mathrm{treated}\;\mathrm{seed}}{\mathrm{Fungal}\;\mathrm{contamination}\;\mathrm{in}\;\mathrm{untreated}\;\mathrm{seed}}\times 100$$

AFB_1_ content was determined in *S. cumini* seeds according to the protocol of Sheijooni Fumani et al. [27]. For this, 5 g grinded seed samples were dissolved in aqueous methanol (8:10 *v*/*v*) in conical flask, succeeded by centrifugation at 11,500× *g* rpm for 8 min. The collected upper layer was added to 6 mL potassium bromide solution (3%) along with 300 µL chloroform. Again, centrifugation of the samples was performed at 10,000× *g* rpm for 10 min, and the lower part was heated in a water bath (78 °C) for 20 min. The remaining material was homogenized into 1 mL methanol and injected in the HPLC system for further analysis. Stock standard solution of AFB_1_ was developed in methanol with limit of detection between 12.5–500 ng 50 mL^−1^. Then, 10 µL of sample was loaded in the C-18 RP-column with a mobile phase of WAM (water, acetonitrile, and methanol) with 1.2 mL min^−1^ flow. Determination of AFB_1_ content was conducted at 365 nm and expressed in terms of µg/kg seed samples.

### 2.14. Effect of CKP-25-EO and CKP-25-Ne on Lipid Peroxidation of Stored S. cumini Seed

Peroxidation of *S. cumini* seeds lipids was measured in terms of malondialdehyde (MDA) content following the method optimized by Das et al. [28] with slight modification. First, 500 mg grinded seed samples (both treated and control) were mixed with 0.375% TBA, 0.25 N HCl, and 15% TCA, followed by heating over a water bath (85 °C) for 15 min. After that, centrifugation of the samples was performed at 10,000× *g* rpm for 10 min, and supernatant was collected for analysis of absorbance at 520 and 600 nm by a UV-Visible spectrophotometer.

### 2.15. Sensory Analysis of CKP-25-EO and CKP-25-Ne Fumigated S. cumini Seeds

Sensory evaluation of *S. cumini* seeds (both treated and control) was conducted for color, texture, flavor, mouth-feel, and overall acceptability [29]. Overall, 10 different panelists from the Banaras Hindu University (BHU) were selected for sensory assessment. The investigation was done following the guidelines and permission of the Animal Care and Ethical Committee of BHU. Panelists signed a written approval regarding the sensitivity of sensory experiments. Each panelist was given 2 min to score different sensory parameters. Scoring of sensory parameters was done by a five-point hedonic scale viz. (10 = exceedingly acceptable, 8 = moderately acceptable, 6 = acceptable, 4 = least acceptable, 2 = unacceptable).

### 2.16. Safety Profile Test of CKP-25-EO and CKP-25-Ne in Mice

For acute oral toxicity assay, three-month-old albino mice were purchased from the Department of Zoology, BHU, and acclimatized for one week. This safety profile testing assay received approval for research ethics from the Ethical Care Committee of BHU. After that, different concentrations of CKP-25-EO and CKP-25-Ne were prepared by dissolving the required amounts into 5% Tween-20 solution. Thereafter, administration of different doses of CKP-25-EO and CKP-25-Ne was performed on mice through a small catheter. Tween-20 and chitosan nanoemulsion were administered to albino mice as control. Mice mortality was observed after 24 h and presented in terms of LD_50_ (50% lethality) value [30]. Probit analysis was used for the calculation of the LD_50_ value [31].

### 2.17. Analysis of Statistical Data

Experiments were performed in triplicate, and results were represented in terms of mean ± standard error format by IBM SPSS Statistics-25 software. One-way-ANOVA was determined by using Tukey’s-b test and significant differences at *p* < 0.05.

## 3. Results and Discussion

### 3.1. Chemical Characterization of CKP-25-EO by GC-MS Analysis

Chemical characterization of CKP-25-EO showed eight compounds constituting 97.66% of total EO (Table 1). Citral (75.67%) was identified as a major ingredient, followed by D-limonene (7.80%), and geranyl acetate (6.21%) in CKP-25-EO. Verma et al. [32] also reported citral as a major ingredient in CKP-25-EO. Variation in the chemical composition of essential oil is depended on the geographical alteration, climatic change, age of plant part, methods applied for essential oil extraction, and chemotypic alteration [33,34]. Hence, before antifungal and AFB_1_ mitigation efficacy testing, analysis of the chemical profile of essential oil is of prime importance.

### 3.2. Preparation of CKP-25-EO Loaded Chitosan Nanoemulsion (CKP-25-Ne)

Chitosan was used for encapsulation of CKP-25-EO due to its biodegradability, biocompatibility, gelation capability, non-toxic nature, and consideration under the GRAS category [35]. Entrapment of CKP-25-EO into chitosan biopolymer was achieved by the ionic gelation method in which +ve charge amino groups (-NH_2_) of chitosan interacted with -ve charge PO_4_^3−^ groups of S-TPP through electrostatic interaction [36]. Different concentrations of CKP-25-EO were used to prepare different ratios of chitosan to CKP-25-EO, which in turn determines the maximum entrapment ability of CKP-25-EO.

### 3.3. Measurement of Efficiency to Load and Encapsulate CKP-25-EO

Percent EE and LC of CKP-25-Ne ranged from 33.80% to 83.15% and 0.29% to 2.21%, respectively (Table 2). Initially, % EE and LC of CKP-25-Ne were increased up to 1:0.8 ratio (chitosan: CKP-25-EO), followed by a major fall in EE and LC values, which may be due to saturation of CKP-25-EO for entrapment into the chitosan nanomatrix as well as feeble binding of CKP-25-EO with chitosan after the threshold level of CKP-25-EO addition to the emulsion system [30,37]. The current finding is consistent with Woranuch and Yoksan [38] reporting a decrease in % EE at a 1:1.25 ratio during entrapment of carvacrol into chitosan nanoparticle. Higher EE and LC demonstrated better protection of the essential oil from the external oxidative environment and also facilitate its application for mitigation of AFB_1_ contamination. As the highest EE and LC were noted at a 1:0.8 ratio (chitosan:CKP-25-EO), this ratio of CKP-25-Ne was therefore used for physico-chemical characterization and biological activity testing throughout the study.

### 3.4. Physico-Chemical Characterization of CKP-25-Ne

#### 3.4.1. Zeta Sizer Study

Average particle size of chitosan nanoemulsion and CKP-25-Ne was 85.41 and 103.56 nm, respectively. The particle size of CKP-25-Ne was found larger as compared to chitosan nanoemulsionic particles (Table 3), which suggested the successful encapsulation of CKP-25-EO as core material inside the chitosan nanobiopolymer. The Zeta potential of chitosan nanoemulsion and CKP-25-Ne was ±41.03 and ±33.91 mV, respectively (Table 3). Both chitosan nanoemulsion and CKP-25-Ne showed positive zeta potential representing the positive surface charge. The surface charge of CKP-25-Ne was decreased due to the armor effect and surface adsorption of CKP-25-EO on chitosan nanoemulsion, leading to the masking of the amino groups of chitosan [39]. The PDI of chitosan nanoemulsion was 0.176, whereas CKP-25-Ne showed the value of 0.162 (Table 3), suggesting homogeneity in particle distribution. Our results are in coherence with the view of Hasani et al. [40] for the decrement in PDI values of chitosan-Hicap nanocapsules after entrapment of lemon EO. Thus, the smaller particle size, homogenous distribution, and greater stability illustrated the suitability for application of CKP-25-Ne as a nano-preservative in food, agricultural, and pharmaceutical industries.

#### 3.4.2. SEM Analysis

The chitosan nanoemulsion and CKP-25-Ne particles exhibited rounded structures with smooth surfaces (Figure 1A,B). The size of the chitosan nanoemulsionic particles ranged between 29.09–91.49 nm, while CKP-25-Ne particles were found in the range of 39.06–107.90 nm. Enhancement in the size of CKP-25-Ne particles as compared to chitosan nanoparticles has been linked with the encompassment of CKP-25-EO as a core material inside the chitosan nanobiopolymer. The present result is consistent with the previous finding of Mohammadi et al. [39] reporting enhancement in the size of chitosan nanoparticles after incorporation of *Cinnamomum zeylanicum* essential oil.

#### 3.4.3. Atomic Force Microscopy (AFM) Analysis

Particle size of CKP-25-Ne through AFM ranged between 67–100 nm (Figure 1C,D). Variation in the size of nanoparticles in SEM and AFM has been associated with the differential sensitivity of the instruments. The current finding is in agreement with the previous finding of Hosseini et al. [41], where the size of oregano essential oil particles embedded in chitosan ranged between 40–80 nm. The sub-cellular size of particles as observed in the present study could help the controlled delivery of CKP-25-EO with better antifungal and antiaflatoxigenic bioactivities.

#### 3.4.4. FTIR Analysis

The FTIR analysis of chitosan powder, chitosan nanoemulsion, CKP-25-EO, and CKP-25-Ne is displayed in Figure 1E. Chitosan showed different peaks at 660 (C–H stretching), 1080 (C=O stretching), 1380 (C–H bending), 1590, 1650 (–NH bending), 2865 (C–H stretching), and 3450 (dimeric –OH stretching) cm^−1^ [42,43]. Chitosan nanoemulsion displayed characteristic peaks at 570, 670 (shifting of 660), 1410 (shifting of 3450), and 3430 (shifting of 1380, 1590), along with the formation of new peaks at 1110 (O–C), 1575 (–N=N–), 1740 (aldehyde), and 2930 (methylene group) cm^−1^. The specific peak at 1250 cm^−1^ explained the binding of the amino functional group of chitosan with the PO_4_^3-^ group of S-TPP [44]. Several peaks at 590 (–OH bending), 840 (nitrate ion), 1000 (C–C), 1230 (–OH stretching), 1380 (methyl C–H bending), 1445 (C–H bending), 1675 (C–H bending), 1740 (aldehyde), 2915 (C–H stretching), and 3400 cm^−1^ (OH stretching) have been shown by CKP-25-EO. CKP-25-Ne exhibited spectral peaks at 570 (C–O stretching), 890 (vinyl C–H out of bending), 1100 (phosphate ion), 1410 (–OH bending), 1560 (aliphatic nitro compound), 2915 (methylene C–H stretching), and 3430 (–NH stretching) cm^−1^ [29], showing a similar peaks arrangement to the chitosan nanoparticle and CKP-25-EO, confirming the successful encapsulation of CKP-25-EO into the chitosan nanomatrix.

#### 3.4.5. XRD Analysis

Chitosan powder exhibited a specific peak at 2θ values 11° and 21°, indicating a high degree of crystallinity (Figure 1F). The diffractogram of chitosan nanoemulsion was broadened between 12–28°, suggesting the destruction in chitosan crystallinity which has been associated with cross-linking of chitosan’s NH_3_^+^ group with S-TPP’s PO_4_^3−^ groups [29]. After encapsulation of CKP-25-EO into the chitosan nanobiopolymer, a greater disarray and shifting of peaks were recorded (Figure 1F), indicating the formation of amorphous structures [28]. The advantage of the chitosan amorphous structure indicated the successful encapsulation of CKP-25-EO into chitosan nanobiopolymer. Moreover, the crystalline nature of native chitosan did not permit it to develop powder or emulsion due to peak stiffness and low adsorption capacity, while the destruction of peak intensity after cross-linking interaction developed amorphous structures which have increased porosity, expanded polymer nanostructural compaction, and improved access to internal sorption sites with delivery applications [45]. The XRD results implied successful incorporation of CKP-25-EO, resulting in possible alteration in chitosan–TPP packing confirmations.

### 3.5. Release Characteristics of CKP-25-EO

In the present investigation, CKP-25-EO showed a biphasic release profile, i.e., initial burst release followed by slow release over a period of 140 h. Figure 2A represents that in the first phase, 27.03% release of CKP-25-EO was observed followed by 22.50% and 12.62% after 12 and 24 h, respectively. After 80 h, the release rate was observed to be 0.23%, suggesting very slow delivery of CKP-25-EO from the chitosan nanoemulsion. The initial burst release of CKP-25-EO might be due to the adsorption at the chitosan surface, and the small size of nanocapsule with more surface area, and fragile interaction between CKP-25-EO and chitosan in the ethanolic PBS solution [46]. However, the later controlled release has been associated with emulsionic expansion, and tight packing of CKP-25-EO inside chitosan [47]—therefore, little by little delivery in PBS solution. The controlled release of CKP-25-EO was mainly attributed through the hydrocarbon portions of the coating layers and the pores contained in the layer [40]. At this time, the dominant discharge system was modified to CKP-25-EO diffusion through the chitosan matrix, which was mainly due to the inefficacy of the buffer solution to decimate the particles, bringing about no extra discharge of essential oil [39]. Sustained delivery of CKP-25-EO facilitates the application of nanoemulsion for practical purposes, with long-term protection of the stored food commodities.

### 3.6. Antifungal and AFB_1_ Suppressing Potentiality of CKP-25-EO and CKP-25-Ne

Complete suppression of growth of AFLHPSc-1 and AFB1 secretion by CKP-25-EO was recorded at 0.4 and 0.35 µL/mL, respectively. The CKP-25-Ne completely inhibited the proliferation of AFLHPSc-1 and AFB1 biosynthesis at 0.08 and 0.07 µL/mL, respectively (Table 4). Both the CKP-25-EO and CKP-25-Ne showed superior antifungal and antiaflatoxigenic activity as compared to chemical disinfectants viz. gallic acid, ascorbic acid, salicylic acid, and butylated hydroxytoluene (BHT). BHT, gallic acid, and ascorbic acid completely inhibited the growth of *Aspergillus flavus* at 10.0 µL/mL. Gallic acid and ascorbic acid caused 100% inhibition of AFB_1_ production at 10.0 µL/mL, while BHT caused 68.61% inhibition of AFB_1_ production by 10.0 µL/mL. Hence, CKP-25-EO showed better activity for inhibition of *A. flavus* growth and AFB_1_ biosynthesis [48]. Better efficacy of CKP-25-Ne for inhibition of fungal growth and AFB_1_ production over CKP-25-EO might be due to synergistic activity of chitosan and CKP-25-EO forming sub-cellular size composite particles, and control release of bioactive ingredients [49]. Variability in antifungal and anti-AFB_1_ activity of CKP-25-Ne has been linked with different pathways involving the inhibition of carbohydrate catabolism and sporulation during AFB_1_ synthesis [31].

### 3.7. Fungitoxic Spectrum of CKP-25-EO and CKP-25-Ne

CKP-25-EO and CKP-25-Ne showed a broad range of fungitoxicity against seven food contaminating molds *viz*. *A. luchuensis, A. niger, A. humicola, P. fellutenum, A. glaucus, P. spinolosum*, and *P. albicans* at 0.4 µL/mL and 0.08 µL/mL (Figure 2B) showing the chances for maximum mitigation of *S. cumini* seeds contamination and AFB_1_ production. Hence, nanoencapsulation strengthens the utilization of CKP-25-Ne for the preservation of *S. cumini* seeds.

### 3.8. Action Mode Related to Antifungal and Antiaflatoxigenic Activity

Ergosterol is a lipid steroid in fungi providing rigidity, integrity, and fluidity to fungal plasma membrane [50]. Ergosterol biosynthesis in the plasma membrane of AFLHPSc-1 was decreased with respect to an increasing concentration of CKP-25-EO. Percent reduction in ergosterol content in AFLHPSc-1 cells was found to be 21.32, 36.31, 53.62, 65.54, 76.66, 84.46, 94.18, and 100% after fumigation with 0.05, 0.1, 0.15, 0.2, 0.25, 0.3, 0.35, 0.4 µL/mL of CKP-25-EO, respectively. CKP-25-Ne showed better activity for the inhibition of the ergosterol biosynthesis (Table 4). Impairment in lanosterol-14α demethylase enzymatic activity may be a possible reason for inhibition of ergosterol biosynthesis in AFLHPSc-1 cells by CKP-25-EO [51]. Decreased biosynthesis of ergosterol by CKP-25-Ne as compared to CKP-25-EO has been linked with the synergistic effect of chitosan and CKP-25-EO in emulsion system with sub-cellular size of particles and targeted delivery.

Fumigation of AFLHPSc-1 cells by CKP-25-EO and CKP-25-Ne at 1/2 MIC, MIC and 2 MIC doses showed an increased efflux of Mg^2+^, K^+^, and Ca^2+^ ions (Figure 2C). A similar trend was observed for losses of 260 nm and 280 nm absorbing materials from AFLHPSc-1 cells after fumigation with CKP-25-EO and CKP-25-Ne (Figure 2D). Leakages of 260 and 280 nm absorbing materials corresponded with losses of nucleic acids and proteins, respectively. Moreover, ions, nucleic acids, and proteins are important components in cells, playing a crucial role in regulating cellular processes during growth and development [52]. Hence, their loss could lead to inhibit the mitochondrial energy dependent AFB_1_ production.

Methylglyoxal (MG) is a cytotoxic component formed as a by-product during glycolysis and acts as AFB1 inducing substrate in *A. flavus* cells as well as increases the generation of reactive oxygen species (ROS) by changing the glutathione level [29]. In the control set, the methylglyoxal level was found to be 1371.52 µM/g FW, while AFLHPSc-1 cells fumigated with CKP-25-EO showed 300.16 and 198.07 µM/g FW at MIC and 2 MIC doses, respectively (Figure 2E). CKP-25-Ne exhibited better activity for inhibition of methylglyoxal biosynthesis (Figure 2E). Retardation of methylglyoxal biosynthesis has been represented as a novel antiaflatoxigenic mechanism of action which might be associated with impairment, and down-regulation in functioning of afl-R and nor-1 genes. The present finding is similar to the report of Upadhyay et al. [7] regarding the inhibition of methylglyoxal biosynthesis by *Cistus ladanifer* essential oil. Better inhibition of methylglyoxal production by CKP-25-Ne has been associated with sub-cellular size with greater surface/volume ratio and better reaction kinetics.

### 3.9. In Silico Modeling of Citral with Nor-1 and Pks-A Proteins of AFB_1_ Biosynthesis

The 3D structure of citral was downloaded from PubChem, and Nor-1 and Pks-A proteins were retrieved from the Uniprot database switched with Phyre 2 server. The Nor-1 and Pks-A proteins were selected due to their important role in the synthesis of AFB1. Nor-1 is an oxidation reduction mediated enzyme that converts the first intermediate product, norsolorinic acid (NOR), into averantin (AVN) in the AFB1 biosynthetic pathway, whereas Pks-A is a polyketide synthase that helps in the synthesis of AFB1 in its initial phase from acetyl Co-A. Citral was interacted with Nor-1 and Pks-A receptor proteins through hydrogen bond at ASN-28, ALA-29, and ARG-159 amino acids, respectively (Figure 3A–E). This hydrogen bonding may stimulate the conformational changes in Nor-1 and Pks-A proteins, causing inhibition of AFB1 synthesis. Our result was presented on the basis of binding energy release during the interaction of citral with the amino acid residue of Nor-1 and Pks-A protein complex (Table 5). A similar finding has been demonstrated by Murugan et al. [53] for effective interaction of *Murrya koenigii* essential oil components, with Ver-1 protein leading to inhibition of AFB_1_ synthesis. The significant binding of citral inside the cavity of Nor-1 and Pks-A protein suggested the synergistic activity during in vitro biological efficacy testing. In conclusion, docking of citral with Nor-1 and Pks-A revealed the molecular mechanism for suppression of AFB1 synthesis in *A. flavus* and developed a green insight for mitigation of AFB_1_ in stored agricultural and food products.

### 3.10. Antioxidant Activity and Phenolic Content of CKP-25-EO and CKP-25-Ne

IC_50_ values of CKP-25-EO through DPPH˙ and ABTS^+^. assay were 13.61 and 12.73 µL/mL, respectively. CKP-25-Ne showed improvement in antioxidant activity (IC_50 DPPH_ = 6.94 µL/mL, IC_50 ABTS_ = 5.40 µL/mL) (Figure 4A,B). Antioxidant activity of CKP-25-EO and CKP-25-Ne for ABTS^·+^ assay (IC_50_ value) was slightly more as compared to DPPH˙ assay because of more sensitivity towards bioactive ingredients of CKP-25-EO. The CKP-25-EO showed better antioxidant activity as compared to synthetic antioxidant like salicylic acid (DPPH _IC50_ = 216 μL/mL). The CKP-25-EO was not able to dissolve in an aqueous solution; thereby, low antioxidant activity was recorded. However, solubility of CKP-25-EO was increased after encapsulation into the chitosan nanoemulsion, which facilitates easy delivery of bioactive ingredients in DPPH and ABTS solution [54], justifying the increment in antioxidant activities.

The phenol content of CKP-25-EO was recorded as 5.52 µg/mg gallic acid equivalent. The phenolic content was found in similar range with essential oils like *Pimpinella anisum* (6.48 µg/mg gallic acid) and *Mentha cardiaca* (7.1 µg/mg gallic acid), which are commonly used as effective fungitoxicant in stored food commodities [25,55]. After encapsulation of CKP-25-EO into chitosan nanoemulsion, better phenolic content was recorded (6.06 µg/mg gallic acid equivalent). Increased phenolic content of CKP-25-Ne might be due to the sub-cellular size of CKP-25-EO facilitating effective gallic acid dialdehyde complex formation in emulsion system [56]. Moreover, an increase in water solubility of phenolic compounds of essential oil and decreased loss of evaporation and reactivity against external environment after encapsulation into chitosan nanobiopolymer may be a possible reason for the increment in phenolic content of CKP-25-Ne [55,57]. Better antioxidant capacity and phenolic content of CKP-25-Ne facilitated the ROS mediated diminution of AFB_1_ contamination in *S. cumini* seeds.

### 3.11. In Situ Antifungal and Anti-AFB_1_ Efficacy of CKP-25-EO and CKP-25-Ne in Stored S. cumini Seeds

During practical application in the real food system, the activity of essential oil is affected by temperature, moisture, pH, oxygen, and chemical composition of commodities. Therefore, in situ experiment is an important parameter for consideration in commercial utilization. In the present investigation, CKP-25-EO at MIC and 2 MIC concentrations caused 78.69 and 84.52% protection against fungal infestation in *S. cumini* seeds, respectively. However, CKP-25-Ne displayed 100% protection against fungal inhabitation at MIC and 2 MIC concentrations (Figure 4C). Our results revealed maximum AFB_1_ content of 27.56 µg kg^−1^ in control *S. cumini* seeds. AFB_1_ content in control *S. cumini* seeds was found to exceed the maximum limits set by the Commissionable rule of Europe (Commission regulation (EU) No 165/2010, 2010), i.e., 10 μg kg^−1^. CKP-25-EO suppressed the AFB_1_ content up to 1.34 and 1.12 µg kg^−1^ at MIC and 2 MIC concentrations, respectively. Complete protection of *S. cumini* seeds against AFB_1_ production was displayed by CKP-25-Ne at MIC and 2 MIC concentrations (Figure 4D). Improved efficiency of CKP-25-Ne for mitigation of AFB_1_ contamination could be linked with the sub-cellular particle size and controlled-release [22], and alteration in regulatory enzymes of carbohydrate metabolism [8]. The findings of the current study are in agreement with the protection of maize seeds by nanoencapsulated *Toddalia asiatica* essential oil fumigation against *A. flavus* over 45 days of storage. However, the present results showed superiority to preserve *S. cumini* seeds against broad spectrum fungal growth and AFB_1_ secretion for one year of storage and extends its application as a green shelf-life enhancer.

### 3.12. Impact of CKP-25-EO and CKP-25-Ne on Lipid Peroxidation in S. cumini Seeds

Lipid peroxidation is the major problem in food and pharmaceutical commodities during storage leading to qualitative and quantitative loss as well as a reduction in nutritional value. MDA, a secondary hydoperoxide component in lipid peroxidation, is indicator of oxidative stress in stored food commodities. During lipid peroxidation, the chemical reaction of ROS with the double bond of poly unsaturated fatty acids causes the production of toxic components in *S. cumini* seeds [58]. In the present investigation, MDA content was very high in the control set (130.09 µM g^−1^ FW). CKP-25-EO caused a reduction in MDA content up to 54.97 and 26.39 µM g^−1^ FW in *S. cumini* seeds at MIC and 2 MIC doses, respectively (Figure 4E). CKP-25-Ne showed superior efficacy for inhibition of MDA biosynthesis after one year of storage in *S. cumini* seeds. The present finding is similar to the investigation of encapsulated *Zanthoxylum bungeanum* essential oil to mitigate the lipid peroxidation in Chinese-style sausage [59]. However, our finding showed superiority to reduce MDA generation by CKP-25-Ne, which has linked the sub-cellular size of particles with sustained delivery and better reactiveness at the site where fungal cells easily oxidize the unsaturated lipids and produce toxic metabolites.

### 3.13. Sensory Evaluation of CKP-25-EO and CKP-25-Ne Fumigated S. cumini Seeds

The sensory results for different parameters like colour, texture, flavor, mouthfeel, and overall acceptability in CKP-25-EO and CKP-25-Ne fumigated *S. cumini* seeds at MIC and 2 MIC doses is presented in Figure 4F. The control *S. cumini* seeds showed unacceptability for flavor, colour, and mouthfeel due to lipid peroxidation causing off-taste and off-odor. The seeds fumigated with CKP-25-EO at 2 MIC dose displayed a somewhat aromatic smell during consumption as reported by panelists that has been linked with absorption of some of the essential oil components by *S. cumini* seeds. To overcome these challenges, the encapsulated CKP-25-EO into chitosan nanoemulsion demonstrated better acceptability for peoples without altering the sensory profiles of *S. cumini* seeds. A better maintenance of sensory parameters by CKP-25-Ne has been linked with the control release profile preventing the development of off-flavor and off-odor. Our result agreed the view of Karimifar et al. [60] for preservation of color and odor of lamb burger patties by the nanoencapsulated *Ziziphora clinopodioides–Rosmarinus officinalis* essential oil.

### 3.14. Acute Oral Toxicity in Mice

Before recommendation of any plant based preservative formulation for commercial purpose, the mammalian safety profile test is necessary to be considered. A lethal dose of CKP-25-EO and CKP-25-Ne in mice was recorded as 10,481.72 and 13,623.91 µL/kg, respectively. Both CKP-25-EO and CKP-25-Ne showed higher LD_50_ value as compared to synthetic preservatives *viz.* benzoic acid (2000–2500 mg/kg), formic acid (700 mg/kg), and acetic acid (3530 mg/kg) [61]. Subsequently, higher LD_50_ of CKP-25-EO and CKP-25-Ne have displayed negligible toxicity on mammalian system. Moreover, chitosan did not show any toxic effect on the mammalian system. CKP-25-Ne showed greater LD_50_ over its unencapsulated form, suggesting a green horizon for its application as an alternative of synthetic preservatives in pharmaceutical, agriculture, and food industries.

## 4. Conclusions

The encompassment of CKP-25-EO into a chitosan nanomatrix enhanced the antifungal activity, AFB_1_ mitigation ability, and free radical quenching potency over the unencapsulated form. The diminution in ergosterol biosynthesis and increased efflux of cellular constituents confirmed the biochemical mechanisms related to antifungal activity. The decreased methylglyoxal production (aflatoxin inducer) strengthened the application of CKP-25-Ne as novel AFB_1_ inhibitor. Moreover, in silico modeling validated the interaction of citral with Nor-1 and Pks-A proteins, causing suppression to AFB1 biosynthesis. In addition, promising in situ antifungal properties, AFB_1_ inhibition, suppression of lipid peroxidation, maintenance of sensory properties, and high mammalian safety strengthen the utilization of CKP-25-Ne as a green agent to inhibit fungal infestation and AFB_1_ contamination along with better practical perspectives as a natural preservative in food, agriculture, and pharmaceutical industries.

## Figures and Tables

**Figure 1 foods-12-00722-f001:**
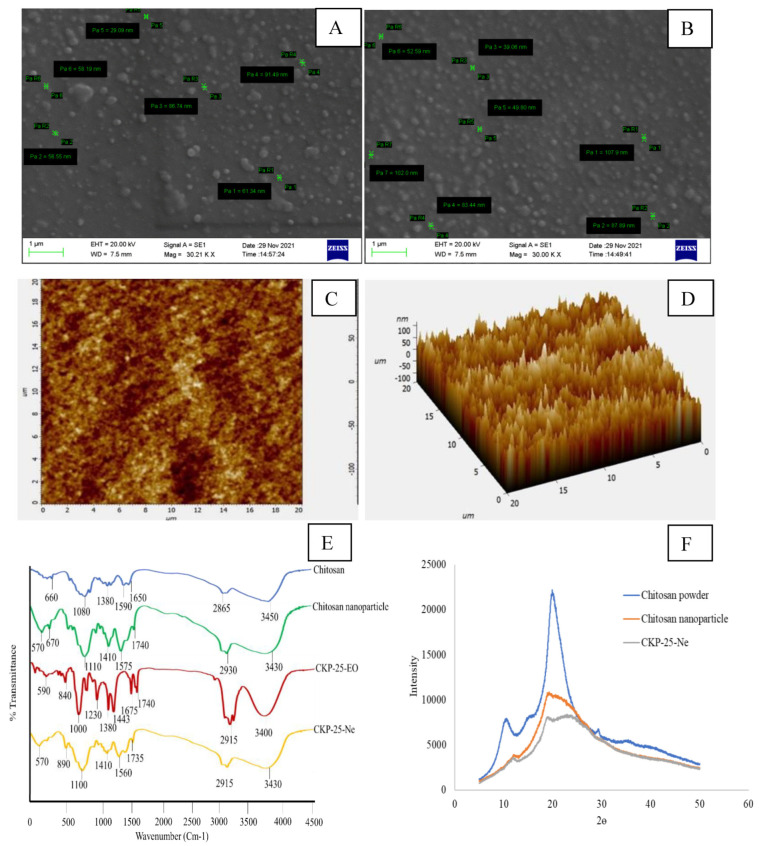
Scanning electron microscope (SEM) image of particles of chitosan nanoemulsion (**A**), SEM image of CKP-25-Ne (**B**), 2D Atomic force microscopic (AFM) image of CKP-25-Ne (**C**), 3D AFM image of CKP-25-Ne (**D**), FTIR analysis of chitosan, chitosan nanoemulsion (particle), CKP-25-Ne, and CKP-25-EO (**E**), XRD analysis of chitosan powder, chitosan nanoemulsion (particle), and CKP-25-Ne (**F**).

**Figure 2 foods-12-00722-f002:**
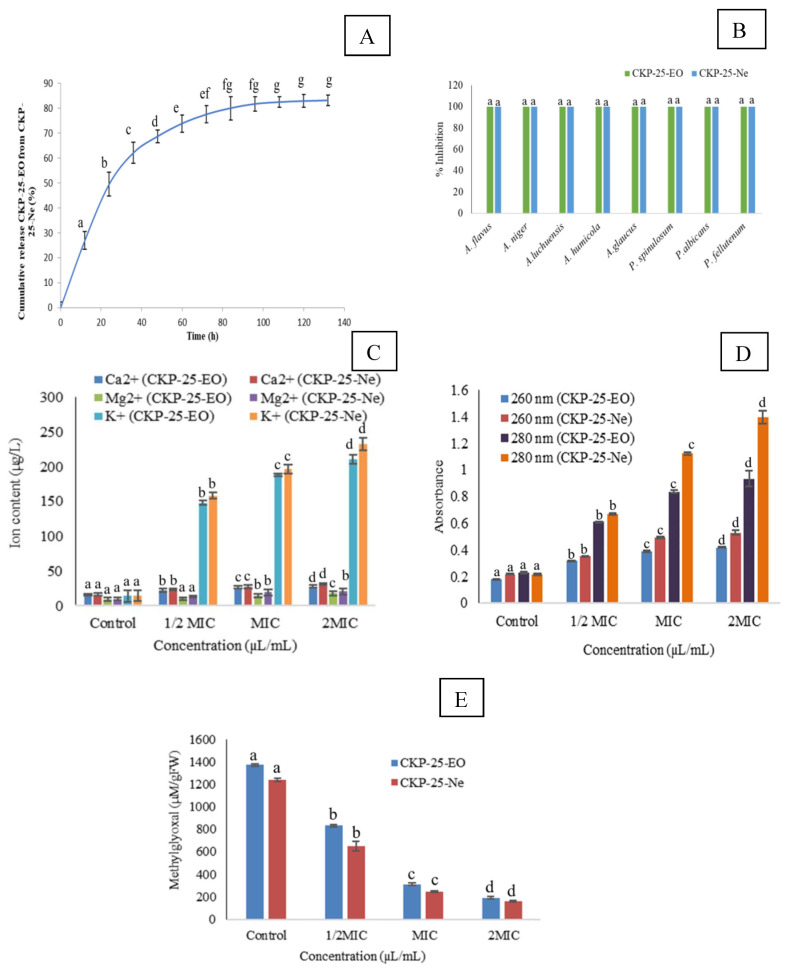
In vitro release profile of CKP-25-Ne (**A**), Fungitoxic spectra of CKP-25-EO and CKP-25-Ne (**B**), Effect of CKP-25-EO and CKP-25-Ne on ions leakage of Ca^2+^, K^+^, Mg^2+^, and 260 nm and 280 nm absorbing materials from AFLHPSc-1 cells (**C**,**D**), Impact of CKP-25-EO and CKP-25-Ne on cellular methylglyoxal (**E**). Note: The letters represent ANOVA (*p* < 0.05).

**Figure 3 foods-12-00722-f003:**
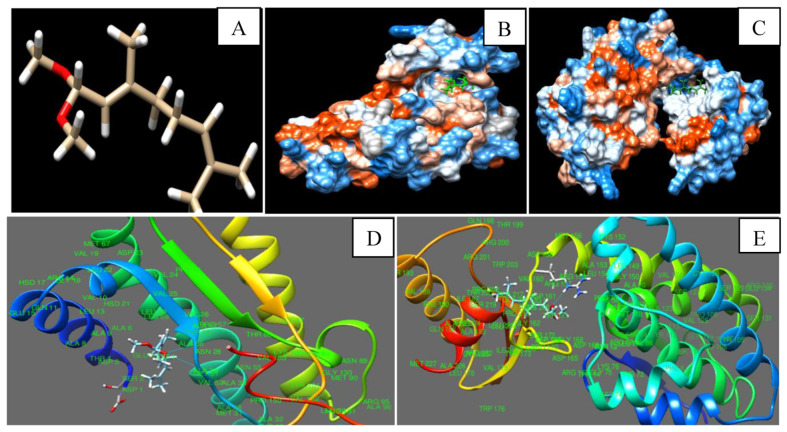
Three-dimensional image of citral (**A**), Interaction of citral with Nor-1 protein (**B**,**D**), Interaction of citral with Pks-A protein (**C**,**E**).

**Figure 4 foods-12-00722-f004:**
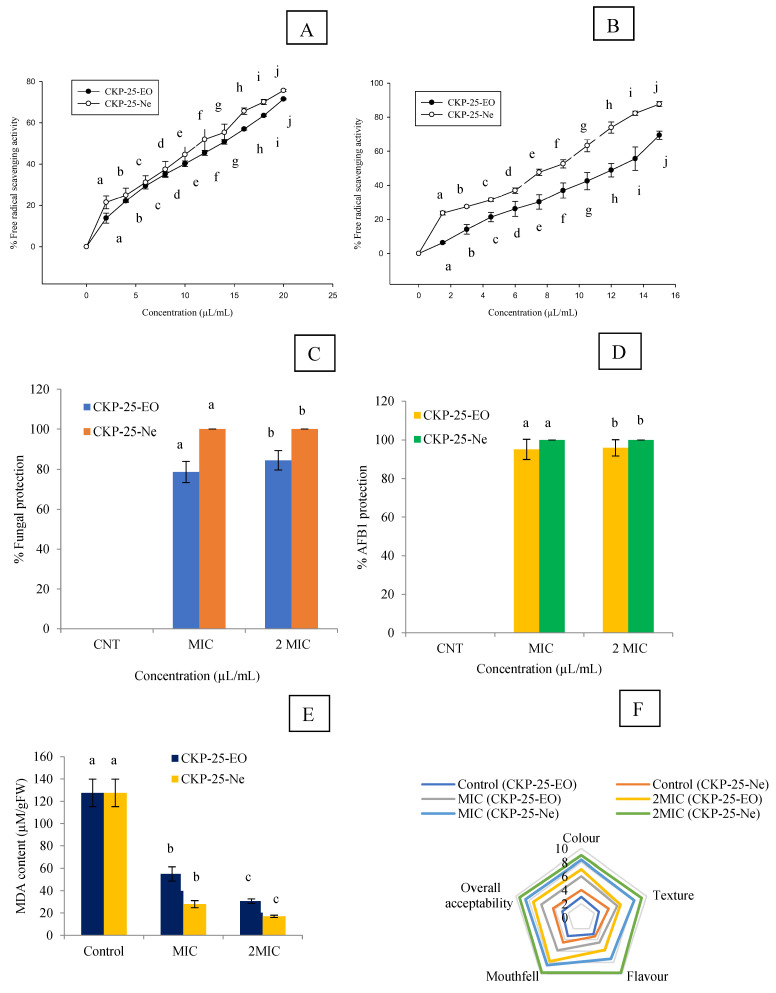
Measurement of antioxidant activity of CKP-25-EO and CKP-25-Ne through DPPH^.+^ and ABTS^.+^ assay (**A**,**B**), Percent protection against fungal infestation and AFB_1_ contamination by CKP-25-EO and CKP-25-Ne in *S. cumini* seeds (**C**,**D**), Effect of CKP-25-EO and CKP-25-Ne on lipid peroxidation in *S. cumini* seeds (**E**), Sensory analysis of CKP-25-EO and CKP-25-Ne fumigated *S*. *cumini* seeds (**F**). Note: The letters represents ANOVA (*p* < 0.05).

**Table 1 foods-12-00722-t001:** GC-MS of CKP-25-EO.

S. No.	RT (min)	% Area	Compounds
1	7.283	2.45	5-Heptene-2-one, 6-methyl-
2	7.392	1.74	β-Myrcene
3	8.131	7.80	D-Limonene
4	9.251	1.21	Linalool
5	10.090	0.95	Citronellal
6	10.228	75.67	Citral
7	11.581	1.64	Geraniol
8	13.354 Total	6.21 97.66	Geranyl acetate

Note: RT = Retention Time.

**Table 2 foods-12-00722-t002:** Encapsulation efficiency and loading capacity of CKP-25-Ne.

S.N.	Chitosan: CKP-25-EO (*w*/*v*)	Encapsulation Efficiency (%)	Loading Capacity (%)
1	1:0	0.00 ± 0.00 ^a^	0.00 ± 0.000 ^a^
2	1:0.2	33.80 ± 0.02 ^b^	0.29 ± 0.34 ^b^
3	1:0.4	44.68 ± 0.23 ^c^	0.59 ± 0.65 ^c^
4	1:0.6	53.07 ± 0.10 ^d^	1.05 ± 0.09 ^d^
5	1:0.8	83.15 ± 0.34 ^e^	2.21 ± 0.78 ^e^
6	1:1	65.88 ± 0.21 ^f^	2.19 ± 0.12 ^f^

Note: Values are mean (*n* = 3) ± SE, the mean followed by same letter in the same column are not significantly different according to ANOVA and Tukey’s multiple comparison tests.

**Table 3 foods-12-00722-t003:** Particles size, zeta potential, and polydispersity index of chitosan nanoemulsion and CKP-25-Ne.

Chitosan: CKP-25 EO Ratio (*w*/*v*)	Average Particle Size (nm)	Zeta Potential (mV)	Polydispersity Index
1: 0 (chitosan nanoemulsion)	85.41± 7.07 ^a^	±41.03 ± 2.01 ^a^	0.176 ± 0.004 ^a^
1: 0.8 (CKP-25-Ne)	103.56 ± 3.86 ^b^	±33.91 ± 1.99 ^b^	0.162 ± 0.009 ^b^

Note: Values are mean (*n* = 3) ± SE, the mean followed by same letter in the same column are not significantly different according to ANOVA and Tukey’s multiple comparison tests.

**Table 4 foods-12-00722-t004:** Antifungal, AFB_1_ inhibitory, and ergosterol reduction potential of CKP-25-EO and CKP-25-Ne.

CKP-25-EO				CKP-25-Ne			
Conc. (µL/mL)	Mycelial Dry Weight (g)	AFB_1_ Content (µg/mL)	% Ergosterol Reduction	Conc. (µL/mL)	Mycelial Dry Weight (g)	AFB_1_ Content (µg/mL)	% Ergosterol Reduction
Control	0.35 ± 0.008 ^a^	32.53 ± 0.083 ^a^	0.00 ± 0.00 ^a^	Control	0.25 ± 0.001 ^a^	26.43 ± 0.602 ^a^	0.00 ± 0.00 ^a^
0.05	0.26 ± 0.001 ^b^	27.00 ± 0.407 ^b^	21.32 ± 4.23 ^b^	0.01	0.21 ± 0.008 ^b^	22.51 ± 0.071 ^b^	23.80 ± 2.15 ^b^
0.1	021 ± 0.002 ^c^	23.04 ± 0.903 ^c^	36.31 ± 5.39 ^c^	0.02	0.17 ± 0.003 ^c^	19.17 ± 0.016 ^c^	38.31 ± 4.11 ^c^
0.15	0.18 ± 0.001 ^cd^	18.94 ± 1.048 ^d^	53.62 ± 3.08 ^d^	0.03	0.15 ± 0.012 ^cd^	15.98 ± 0.708 ^d^	54.24 ± 7.87 ^d^
0.2	0.14 ± 0.006 ^de^	15.17 ± 1.009 ^e^	65.54 ± 3.20 ^de^	0.04	0.12 ± 0.008 ^de^	13.69 ± 0.805 ^d^	68.32 ± 3.08 ^e^
0.25	0.11 ± 0.001 ^ef^	11.40 ± 0.407 ^f^	76.66 ± 5.67 ^ef^	0.05	0.09 ± 0.003 ^ef^	10.25 ± 1.009 ^e^	79.36 ± 1.30 ^ef^
0.3	0.07 ± 0.008 ^fg^	7.39 ± 0.074 ^g^	84.46 ± 4.85 ^fg^	0.06	0.07 ± 0.005 ^f^	7.06 ± 0.806 ^f^	83.77 ± 0.95 ^fg^
0.35	0.04 ± 0.005 ^gh^	0.00 ± 0.00 ^h^	94.18 ± 1.78 ^g^	0.07	0.03 ± 0.005 ^g^	0.00 ± 0.00 ^g^	95.18 ± 1.60 ^gh^
0.4	0.00 ± 0.00 ^h^	0.00 ± 0.00 ^h^	100 ± 0.00 ^g^	0.08	0.00 ± 0.00 ^h^	0.00 ± 0.00 ^g^	100 ± 0.00 ^h^

Note: Values are mean (*n* = 3) ± SE, the mean followed by same letter in the same column are not significantly different according to ANOVA and Tukey’s multiple comparison tests.

**Table 5 foods-12-00722-t005:** Binding energy and hydrogen bond for interaction of Nor-1 and Pks-A with citral.

Essential Oil Component	Receptor Protein	Hydrogen Bonding Amino Acids	Bond Length (Å)	Binding Energy (Kcal/mol)
Citral	Nor-1	ASN 28	2.61	−6.21
		ALA 29	2.54	−6.48
	Pks-A	ARG 159	2.60	−6.67

## Data Availability

Data is contained within the article.

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
