# Peer review of "Encapsulation of Cymbopogon khasiana × Cymbopogon pendulus Essential Oil (CKP-25) in Chitosan Nanoemulsion as a Green and Novel Strategy for Mitigation of Fungal Association and Aflatoxin B1 Contamination in Food System"

_foods, 2023, doi:10.3390/foods12040722_

Round 1

Reviewer 1 Report

The manuscritpt entitled: "Encapsulation of Cymbopogon khasiana × Cymbopogon pendulus essential oil (CKP-25) in chitosan nanoemulsion as a green and novel strategy for mitigation of fungal association and Aflatoxin B1 contamination in food system" present very complex study.

The methodology is appropriate with wide range of the methodology to support the scientific research.

Article is written very clear with the easy reading results.

Only minor corrections are recommended:

Chapter 2.2 – a brief description of the antifungal method should be added

 Line 121: correct „GC-MS headspace &amp

Author Response

Manuscript ID: foods-2122784

Title: Encapsulation of Cymbopogon khasiana × Cymbopogon pendulus essential oil (CKP-25) in chitosan nanoemulsion as a green and novel strategy for mitigation of fungal association and Aflatoxin B1 contamination in food system

Comments to Author

Reviewer #1: 

The methodology is appropriate with wide range of the methodology to support the scientific research. Article is written very clear with the easy reading results. Only minor corrections are recommended:

Query 1. Chapter 2.2 A brief description of the antifungal method should be added

Response: Thanks for the suggestion. Chapter 2.2 indicated the fungal strains used in the present investigation. Description of antifungal method has been mentioned in the section 2.8 of the revised manuscript.

Query 2. Line 121-Correct „GC-MS headspace & amp.

Response: Now, the sentence has been corrected in the revised manuscript.

Note: All the correction made in revised manuscript is highlighted with red font

A K Dwivedy

Reviewer 2 Report

the manuscript is a very interesting research in the formulation of essential oils to avoid microbial contamination of grains for human consumption.

The choice of chitosan is a very good one, as this is used for drug delivery, approved by the FDA and non-toxic, even more, it can be degraded by enzymes.

I would recommend:

In introduction: 1) add more citations, this section is providing a lot of information and many statements belong to only 1 paper, certainly it needs more sources; 2) add some information about the mode of release for nano-encapsulations, the way they work, etc.

Materials and methods: 1) check the decimal figures, it is 0.60 rather than 0.6. 2) Add space for (XRD)observation, same for pH =7.4. 3) Replace 1/2h by 30 minutes. 4) Replace bluish for light blue. 5) check is space is needed in all the temperature parts 27±2ºC, also, in some cases the degree symbol is degree symbol and in others it is number symbol. 6) the line in 297 is different font, this is generally the same case for all formulas. 7) the dilution 10-4 could be replaced by 1/10,000 that is more in line for manuscripts. 8) line 315, replace "the left material" by "the remaining material". 9) there is mention of ethical approval for human test subjects but nothing related to the test mice, has this been done?

Results and discussions 1) the results have been compared to literature, for the extracted CKP-25, has this been compared to a commercial source to see if it is similar? 2) table 2, is there a parameter to add for the expected efficiency or loading capacity to establish what would be expected or considered appropriate? 3) table 5 is broken, please check if it can be placed properly. 4) IC50, the 50 needs to be in subscript. 5) line 588, the x is in subscript, this needs to be corrected. 6) the antioxidant activity and phenolic content should be compared to the expected one, before encapsulation, has this been done? there is a mention of increase phenolic content, is this to the expected one or another one? it is a bit confusing, explain more the concept of subcellular size.  7) has it been investigated the antimicrobial properties in the grain in function of time? to see for degradation, half life, etc?

References, please check as some numbers are in italics and some are not.

Figure 2, it is confusing the letters in the figures, the colours are ok for explanation but then the letters are not. Same for figure 4.

Author Response

Manuscript ID: foods-2122784

Title: Encapsulation of Cymbopogon khasiana × Cymbopogon pendulus essential oil (CKP-25) in chitosan nanoemulsion as a green and novel strategy for mitigation of fungal association and Aflatoxin B1 contamination in food system

Comments to Author

Reviewer #2:

Query 1. In Introduction add more citations, this section is providing a lot of information and many statements belong to only 1 paper, certainly it needs more sources.

Response: Thanks for the suggestion. Now, we have incorporated additional citations from various sources with emphasis on recent investigations.

Query 2. Add some information about the mode of release for nano-encapsulations, the way they work, etc.

Response: Now, we have incorporated the mode of release of essential oil from nanoemulsion in Introduction and section 3.5. of the revised manuscript.

Query 3. Check the decimal figures, it is 0.60 rather than 0.6.

Response: We apologize for the typing mistake. It has been corrected as 0.06 instead of 0.6 in the revised manuscript. 

Query 4. Add space for (XRD)observation, same for pH =7.4.

Response: Now, we have provided space for XRD observation and pH in the revised manuscript.

Query 5. Replace 1/2h by 30 minutes.

Response: Now, ½ h has been replaced by 30 minutes in the revised manuscript.

Query 6. Replace bluish for light blue.

Response: “Bluish” has now been corrected to “light blue” in the revised manuscript.

Query 7. Check is space is needed in all the temperature parts 27±2°C, also, in some cases the degree symbol is degree symbol and in others it is number symbol.

Response: Now, we have checked the temperature parts and degree symbols throughout the manuscript.

Query 8. The line in 297 is different font, this is generally the same case for all formulas.

Response: Now, we have checked the font size and same case was used for all the formulas.

Query 9. The dilution 10-4 could be replaced by 1/10,000 that is more in line for manuscripts.

Response: Now, we have replaced 10-4 with 1/10,000 in the revised manuscript.

Query 10. Line 315, replace "the left material" by "the remaining material".

Response: Now, “the left material” has been replaced by “the remaining material” in the revised manuscript.

Query 11. There is mention of ethical approval for human test subjects but nothing related to the test mice, has this been done?

Response: Ethical approval was taken from Ethical care Committee of Banaras Hindu University for safety profile assessment in mice and sensory analysis of test samples involving human panelists.

Query 12. The results have been compared to literature, for the extracted CKP-25, has this been compared to a commercial source to see if it is similar?

Response: In the present investigation, we have compared the antifungal and antiaflatoxigenic activity of CKP-25-EO with commonly used food preservatives like butylated hydroxyl toluene (BHT), gallic acid, and ascorbic acid. BHT, gallic acid, and ascorbic acid completely inhibited the growth of Aspergillus flavus at 10.0 µL/mL. Gallic acid and ascorbic acid caused 100 % inhibition of AFB1 production at 10.0 µL/mL, while BHT caused 68.61 % inhibition of AFB1 production by 10.0 µL/mL. Hence, CKP-25-EO showed better activity for inhibition of A. flavus growth and AFB1 biosynthesis (Refer to section 3.6.).

Query 13. Table 2, is there a parameter to add for the expected efficiency or loading capacity to establish what would be expected or considered appropriate?

Response: In the present investigation it is not needed to add a parameter for the expected efficiency or loading capacity because the loading capacity and encapsulation efficiency just denoted the per unit loading or encapsulation of CKP-25-EO into chitosan nanoemulsion. Since, LC and EE varies from material to material. Therefore, we screen the ratio of chitosan to essential oil with maximum loading and encapsulation efficiency. Hence, the nanoemulsion with highest LC and EE has been selected for detailed studies.

Query 14. Table 5 is broken, please check if it can be placed properly.

Response: Now, the table 5 has been corrected in the revised manuscript.

Query 15. IC50, the 50 needs to be in subscript.

Response: In IC50, the 50 has now been placed in subscript.

Query 16. Line 588, the x is in subscript, this needs to be corrected.

Response: The correction has now been incorporated in the revised manuscript.

Query 17. The antioxidant activity and phenolic content should be compared to the expected one, before encapsulation, has this been done? there is a mention of increase phenolic content, is this to the expected one or another one? it is a bit confusing, explain more the concept of subcellular size. 

Response: Phenolic content is responsible for antioxidant activity hence; we have measured it. The CKP-25-EO showed better antioxidant activity as compared to synthetic antioxidant like salicylic acid (DPPH IC50 = 216 μL/mL). We have not compared the phenolic content of CKP-25-EO with any synthetic antioxidants, however, the phenolic content was found in similar range with essential oils like Pimpinella anisum (6.48 µg/mg gallic acid) and Mentha cardiaca (7.1 µg/mg gallic acid) which are commonly used as effective fungitoxicant in stored food commodities (Das et al., 2021; Dwivedy et al., 2017).

The phenolic content of CKP-25-EO was increased after encapsulation into chitosan biopolymer. The reason for possible increment in phenolic content of CKP-25-Ne was increase in water solubility of phenolic compounds of essential oil and decreased loss of evaporation and reactivity against external environment (Desai & Park, 2005; Zuidam & Heinrich, 2010).

Dwivedy, A. K., Prakash, B., Chanotiya, C. S., Bisht, D., & Dubey, N. K. (2017). Chemically characterized Mentha cardiaca L. essential oil as plant based preservative in view of efficacy against biodeteriorating fungi of dry fruits, aflatoxin secretion, lipid peroxidation and safety profile assessment. Food and chemical toxicology106, 175-184.

Das, S., Kumar Singh, V., Kumar Dwivedy, A., Kumar Chaudhari, A., Upadhyay, N., Singh, A., & Dubey, N. K. (2020). Assessment of chemically characterised Myristica fragrans essential oil against fungi contaminating stored scented rice and its mode of action as novel aflatoxin inhibitor. Natural product research34(11), 1611-1615.

Query 18. Has it been investigated the antimicrobial properties in the grain in function of time? to see for degradation, half life, etc?

Response: We have investigated the antifungal activity of CKP-25-EO and CKP-25-Ne in Syzygium cumini seeds to demonstrate the in situ efficacy in food system after 1 year of storage period. The effective inhibition of fungal infestation and AFB1 contamination by CKP-25-EO and CKP-25-Ne suggested the controlled delivery and long-term maintenance of essential oil activity in the food system.  We have not seen the degradation half-life for the present studies, but, we are planning to observe the temporal change in the degradation of the stored food commodities in our future projects.

Query 19. References, please check as some numbers are in italics and some are not.

Response: Now, we have thoroughly checked the references in the revised manuscript.

Query 20. Figure 2, it is confusing the letters in the figures, the colours are ok for explanation but then the letters are not. Same for figure 4.

Response: In the Fig. 2 and Fig. 4 the letters represent significant differences at p value < 0.05 according to ANOVA and Tukey’s multiple comparison tests.

Note: All the corrections made in the revised manuscript are highlighted with red font.

A. K. Dwivedy

Reviewer 3 Report

Dear Authors,

This manuscript written by Jitendra Prasad et al. is a very excellent research on. " Encapsulation of Cymbopogon khasiana × Cymbopogon pendulus essential oil (CKP-25) in chitosan nanoemulsion as a green and novel strategy for mitigation of fungal association and Aflatoxin B1 contamination in food system”. I would like to say that it is excellent research, but, the study needs minör revision.

1. Antioxidant activity values should be stated in the following sentence in the abstract section.

“The CKP-25-Ne displayed enhanced anti20 fungal (0.08 µL/mL), antiaflatoxigenic (0.07 µL/mL), and antioxidant activities”

2. In the Reagents and chemicals section, the properties of some chemicals should be specified (for example, methanol ???) and the clear name should be written at first use, for example DPPH???

3.In the introduction section, the reasons for the need for encapsulation of chitosan should also be added.

4. Literature citations in the introduction section are quite inadequate. More findings are needed regarding related studies. The following article may help you, Moreover, There is no refere for analysis result of FTIR for chitosan

5. Please provide reference for the ""Extraction of Lemongrass (CKP-25) essential oil"" process

6. As you know, the existence of nanoparticles is proven by TEM. I would prefer TEM analysis to SEM analysis

7.plagiarism ratio is really good

8.Minor grammatical errors in English need to be corrected.

9. Please refer to the formulas below

Encapsulation efficiency (EE) (%) = Total amount of CKP−25 into nanoemulsion

Initial amount of CKP−25−EO

154×100

Loading capacity (LC) (%) Also Release of CKP-25-EO (%).

10. I recommend you to use ethanol for antioxidant activities in your future studies.

11. It is nice to mention the amorphous structure obtained in the XRD analysis results in the following sentence, but the advantages of obtaining this amorphous structure need to be mentioned.

After encapsulation of CKP-25-

467 EO into chitosan nanobiopolymer greater disarray and shifting of peaks were

468 recorded (Fig. 1F), indicating the formation of amorphous structures [17]

12. In Figure 2 and 4, the concentration values on the x-axis should be numerically indicated on the data label.

Author Response

Manuscript ID: foods-2122784

Title: Encapsulation of Cymbopogon khasiana × Cymbopogon pendulus essential oil (CKP-25) in chitosan nanoemulsion as a green and novel strategy for mitigation of fungal association and Aflatoxin B1 contamination in food system

Comments to Author

Reviewer #3:

Query 1. Antioxidant activity values should be stated in the following sentence in the abstract section.

“The CKP-25-Ne displayed enhanced anti20 fungal (0.08 µL/mL), antiaflatoxigenic (0.07 µL/mL), and antioxidant activities”

Response: Now, we have incorporated the antioxidant value in the mentioned sentence.

Query 2. In the Reagents and chemicals section, the properties of some chemicals should be specified (for example, methanol ???) and the clear name should be written at first use, for example DPPH???

Response: Suggestion incorporated in the revised manuscript.

Query 3. In the introduction section, the reasons for the need for encapsulation of chitosan should also be added.

Response: The need for encapsulation of essential oil in chitosan has now been incorporated in the revised manuscript.

Query 4. Literature citations in the introduction section are quite inadequate. More findings are needed regarding related studies. The following article may help you, Moreover, There is no refere for analysis result of FTIR for chitosan.

Response: The introduction section has now been reorganized with emphasis on adequate recent references in the revised manuscript. Reference for FTIR analysis of chitosan has now been incorporated in the revised manuscript.

Query 5. Please provide reference for the ""Extraction of Lemongrass (CKP-25) essential oil"" process.

Response: Now, we have provided a reference for "Extraction of Lemongrass (CKP-25) essential oil" process in the revised manuscript.

Query 6. As you know, the existence of nanoparticles is proven by TEM. I would prefer TEM analysis to SEM analysis.

Response: Scanning electron microscopy reveals the size as well as the surface characteristics of nanoparticles. Hence, we have used SEM analysis to determine the size and morphology of the nanoparticles. However, we will incorporate TEM analysis in our future investigations.

Query 7. Plagiarism ratio is really good.

Response: We have further reduced the plagiarism throughout the manuscript.

Query 8. Minor grammatical errors in English need to be corrected.

Response: The grammatical errors have now been corrected in the revised manuscript.

Query 9. Please refer to the formulas below

Encapsulation efficiency (EE) (%) = Total amount of CKP−25 into nanoemulsion

Initial amount of CKP−25−EO

154×100

Loading capacity (LC) (%) Also Release of CKP-25-EO (%).

Response: We are unable to understand what the learned reviewer wants to say in this comment.

Query 10. I recommend you to use ethanol for antioxidant activities in your future studies.

Response: Thanks for the suggestion. We will use ethanol for antioxidant activities in our future studies.

Query. 11.  It is nice to mention the amorphous structure obtained in the XRD analysis results in the following sentence, but the advantages of obtaining this amorphous structure need to be mentioned.

After encapsulation of CKP-25-EO into chitosan nanobiopolymer greater disarray and shifting of peaks were recorded (Fig. 1F), indicating the formation of amorphous structures [17]

Response: The amorphous structure of the chitosan is the result of loss of crystallinity in the native chitosan. The amorphous structure of CKP-25-Ne confirmed the successful entrapment of CKP-25-EO into chitosan nanoemulsion. This amorphous structure of chitosan was developed due to destruction of specific chitosan peaks at 11 and 21°. Moreover, the crystalline nature of native chitosan did not permit it to develop powder or emulsion due to peak stiffness and low adsorption capacity while the destruction of peak intensity after crosslinking interaction developed amorphous structures which have increased porosity, expanded polymer nanostructural compaction and improved access to internal sorption sites with delivery applications (Qu and Luo, 2020).

Qu, B., & Luo, Y. (2020). Chitosan-based hydrogel beads: Preparations, modifications and applications in food and agriculture sectors–A review. International journal of biological macromolecules152, 437-448.

Query 12. In Figure 2 and 4, the concentration values on the x-axis should be numerically indicated on the data label.

Response: We are in agreement with the learned reviewer, but, in the case of figures 2 and 4 the individual concentration values have not been mentioned only to enhance the clarity of individual points (due to a large number of data set) and make the statistical figures more visible. Moreover, the respective concentration can be easily obtained from respective axes.

Note: All the corrections made in the revised manuscript are highlighted with red font.

A. K. Dwivedy
